# Advanced Extraction Techniques and Physicochemical Properties of Carrageenan from a Novel *Kappaphycus alvarezii* Cultivar

**DOI:** 10.3390/md22110491

**Published:** 2024-10-31

**Authors:** Madalena Mendes, João Cotas, Irene B. Gutiérrez, Ana M. M. Gonçalves, Alan T. Critchley, Lourie Ann R. Hinaloc, Michael Y. Roleda, Leonel Pereira

**Affiliations:** 1CFE—Centre for Functional Ecology: Science for People & Planet, Marine Resources, Conservation and Technology—Marine Algae Lab, Department of Life Sciences, University of Coimbra, 3000-456 Coimbra, Portugal; 2018283963@student.uc.pt (M.M.); jcotas@uc.pt (J.C.); amgoncalves@uc.pt (A.M.M.G.); 2Marine and Environmental Sciences Centre (MARE), Department of Life Sciences, University of Coimbra, 3000-456 Coimbra, Portugal; ibgutierrez@uc.pt; 3Department of Biology and CESAM, University of Aveiro, 3810-193 Aveiro, Portugal; 4Verschuren Centre for Sustainability in Energy and Environment, Sydney, NC B1M 1A2, Canada; alan.critchley2016@gmail.com; 5The Marine Science Institute, College of Science, University of the Philippines Diliman (UP-MSI), Quezon City 1101, Philippines; lrhinaloc@msi.upd.edu.ph (L.A.R.H.); myroleda@up.edu.ph (M.Y.R.); 6Bolinao Marine Laboratory, University of the Philippines Diliman (UP-MSI), Luciente I, Bolinao 2406, Philippines

**Keywords:** marine polysaccharides, kappa-carrageenan, alkaline treatment, ultrasound-assisted extraction, supercritical fluid extraction, semi-refined carrageenan (SRC), food safety

## Abstract

Carrageenans are valuable marine polysaccharides derived from specific species of red seaweed (Rhodophyta) widely used as thickening and stabilizing agents across various industries. *Kappaphycus alvarezii*, predominantly cultivated in tropical countries, is the primary source of kappa-carrageenan. Traditional industrial extraction methods involve alkaline treatment for up to three hours followed by heating, which is inefficient and generates substantial waste. Thus, developing improved extraction techniques would be helpful for enhancing efficiency and reducing environmental impacts, solvent costs, energy consumption, and the required processing time. In this study, we explored innovative extraction methods, such as ultrasound-assisted extraction (UAE) and supercritical water extraction (SFE), together with other extraction methods to produce kappa-carrageenan from a new strain of *K. alvarezii* from the Philippines. FTIR-ATR spectroscopy was employed to characterize the structure of the different carrageenan fractions. We also examined the physicochemical properties of isolated phycocolloids, including viscosity, and the content of fatty acids, proteins, and carbohydrates. For refined carrageenan (RC), both the traditional extraction method and the UAE method used 1 M NaOH. Additionally, UAE (8% KOH) was employed to produce semi-refined carrageenan (SRC). UAE (8% KOH) produced a high yield of carrageenan, in half the extraction time (extraction yield: 76.70 ± 1.44), and improved carrageenan viscosity (658.7 cP), making this technique highly promising for industrial scaling up. On the other hand, SFE also yielded a significant amount of carrageenan, but the resulting product had the lowest viscosity and an acidic pH, posing safety concerns as classified by the EFSA’s re-evaluation of carrageenan as a food additive.

## 1. Introduction

Traditional methods for extracting polysaccharides from seaweed are labor-intensive and require substantial quantities of chemicals, water, and energy. Hot water extraction (HWE) is one of the most commonly used techniques, but it demands extended extraction times at high temperatures and yields low returns, including low selectivity and extraction efficiency. This method typically produces “native” (i.e., non-modified) polysaccharides [1].

Green industries are emerging to contribute to the reduction in carbon footprints and the consequent climate change impacts by scaling down industrial chemical usage and processing time while enhancing the quality and quantity of extracted seaweed polysaccharides. This creates a beneficial scenario both environmentally and economically. Some of these innovative industrial testing techniques include the following: microwave-assisted extraction (MAE); enzymatic-assisted extraction (EAE); ultrasound-assisted extraction (UAE), which operates at lower temperatures and yields higher outputs [2]; and green solvent extraction methods like subcritical water extraction (SWE) and ionic liquid extraction. Other advanced methods include supercritical fluid extraction (SFE) [3], reactive extrusion, and photo-bleaching. 

Seaweeds from the family Solieriaceae (Gigartinales, Rhodophyta), specifically from the genera *Kappaphycus* and *Eucheuma*, are collectively known as eucheumatoids. Eucheumatoids do not produce pure forms of κ- or ι-carrageenans but instead generate a variety of hybrid structures. These macroalgae produce various types of carrageenans, primarily kappa and iota. The domestication of these seaweeds began in the 1960s when harvesting wild populations could no longer meet the increasing global demand for carrageenans. Carrageenans are polysaccharides composed of D-galactose and 3,6-anhydro-D-galactose sulfate. Their chemical structure is heterogeneous and classified based on the number and position of sulfate esters and the location of the 3,6-anhydro-bridge in (1→4)-linked galactopyranose molecules. The industrially significant types extracted from various eucheumatoids are kappa (κ)-, iota (ι)-, and lambda (λ)-carrageenans [4]. Kappa-carrageenan features a sulfate group on the O-4 position of every third galactosyl unit in its repeating dimer. Its structure consists of a right-handed double helix formed by parallel chains [4]. To chemically stabilize carrageenan, there is a need for the alkali modification of carrageenan in industrial extraction to enhance polysaccharide extraction yield, boost gel strength, and increase the product’s reactivity with proteins [5]. 

Carrageenans can be classified based on their production method into refined carrageenan (RC) and semi-refined carrageenan (SRC). RC, recognized as food-grade carrageenan, is labeled with the EU additive number E407, whereas SRC is E407a [5]. The European Food Safety Authority (EFSA) considers both refined carrageenan (RC) and semi-refined carrageenan (SRC) to be safe, with no evidence of adverse effects in humans, when consumed in the amounts necessary to achieve the desired food texture [6]. According to the Commission Regulation (EU), the purity criteria for refined carrageenan (RC) and semi-refined carrageenan (SRC) include parameters such as appearance, viscosity, sulfate content, ash content, pH, moisture, solubility, heavy metals/metalloids (arsenic, lead, mercury, and cadmium), and molecular weight. Carrageenans are defined as having an average molecular weight ranging from 200 to 800 kDa [7]. Carrageenan preparations exhibit high polydispersity, with a small fraction of lower-molecular-weight polymeric chains (20–50 kDa) naturally present in all samples [6]. However, under conditions of extensive hydrolysis, at low pH (<1.3) and high temperatures (>80 °C) for extended periods, a non-naturally occurring fraction of degraded carrageenan can form. This degraded fraction has a weight average molecular weight of 10–20 kDa and lacks texturizing properties; it has also been called poligeenan [6]. Studies conducted in rodents have shown that consuming high amounts of poligeenan is associated with the development of ulcerative colitis [8]. Therefore, the number of low-molecular-weight polysaccharide chains in food-grade carrageenan is regulated. Although there is currently no validated testing method for quantifying this low-molecular-weight fraction in carrageenan samples, viscosity measurements can differentiate between carrageenans and poligeenans. A viscosity of 5 mPa·s (for a 1.5% solution at 75 °C) corresponds to a sample with a molecular weight of approximately 100–150 kDa [7].

In the production of RC, polysaccharides are dissolved in hot alkaline solutions, such as sodium hydroxide (NaOH), at temperatures between 95 and 110 °C. Insoluble compounds are then removed through filtration. The solution is concentrated, and carrageenan is purified by precipitating it with alcohol, a method used for purifying all types of carrageenan [9]. Following this step, the biomass is dried and then ground into a powder.

In RC production, the hydrocolloid is extracted from the seaweed matrix. In contrast, SRC processing involves treating seaweed thalli (dried seaweed) with aqueous potassium hydroxide (KOH) at 75–80 °C for 2 h to dissolve and remove soluble compounds other than carrageenans, such as salts, soluble sugars, and proteins [10]. The hydroxide in the reagent interacts with the seaweed polymers, reducing the sulfate content in carrageenan while increasing the amount of 3,6-anhydrogalactose. The potassium component of the reagent reacts with carrageenan to form a gel, which prevents it from dissolving in the hot solution. Soluble carbohydrates, proteins, fats, and salts are removed when the solution is drained, and the residue is rinsed multiple times. This washing step is designed to remove the processing alkali and any other water-soluble compounds. The alkali-treated seaweed (ATC) is then dried, cut, and ground into a powder, known as SRC or seaweed flour [10]. 

RC, often referred to as raw carrageenan [5], is known for its higher quality and clear appearance, making it suitable for a wide range of applications. However, SRC production is more economical than RC extraction as it avoids the costs associated with carrageenan precipitation and solvent recovery. Although SRC is generally of lower quality and less suitable for human food applications, it is increasingly used as a cost-effective extender in food products such as processed meat. SRC is primarily employed in pet food production due to its slightly cloudy and colored appearance and may have a higher bacterial count, which is nullified during meat processing [11].

*Kappaphycus alvarezii* is a commercially valuable red macroalga highly sought after for its cell wall polysaccharide, κ-carrageenan [12]. Once harvested, these seaweeds yield a relatively high polysaccharide content, e.g., approximately 40–60% on a dry weight basis [5]. *K. alvarezii* is preferred by industrial processors for producing both SRC and RC.

Ironically, even after nearly six decades, the sex and ploidy of commercially cultivated *Kappaphycus* species remain entirely understudied. Furthermore, potential differences in growth rates, biochemistry, and rheological properties between vegetative and reproductive thalli and amongst different life history stages (i.e., male and female haploid gametophytes and diploid tetrasporophytes) are rarely considered, despite their possible significant economic impacts. For example, in other commercially important taxa, such as *Gelidiella acerosa*, agar yield and quality vary between vegetative and reproductive fronds [13]. In *Chondrus crispus*, the carrageenan composition varies significantly between life stages: the diploid tetrasporophyte primarily produces lambda-carrageenan, while the haploid gametophyte mainly produces kappa- and iota-carrageenans [4].

Carrageenans have been extracted on an industrial scale primarily from farmed, highly selected tropical eucheumatoids using a conventional method. This process involves exposing seaweed biomass to high temperatures for up to 3 h in an alkaline environment. This traditional method could be improved by employing novel technologies that offer higher extraction yields, better properties, reduced extraction time, and lower production costs.

In this study, native (NE), conventional (CE), and advanced extraction technologies, e.g., ultrasound-assisted extraction (UAE) and supercritical fluid extraction (SFE), were tested to quantify their effects on the yield and properties of extracted carrageenan—a regulated food additive product [2]. Aqueous UAE methods have previously shown that an increased extraction yield in the red alga *Hypnea musciformis* caused the disruption of cell walls, decreased particle size, and increased the mass transfer of cell contents [14]. In the SFE method, temperature and pressure were co-applied to quickly permeate into seaweed thalli in order to produce pure extracts. The efficiency of SFE, on the other hand, is dependent on the water content of the starting seaweed material, as well as the ratio of solvent, the flow rate, temperature, pressure, and the biomass particle size [3]. SFE has several benefits, including the use of environmentally friendly solvents (i.e., water), speedy extraction, and high-quality outputs [1]. However, the above studies lack product quality assurance which substantiate and guarantee the applications of the various extracts as human food additives. It is essential that such products produced using novel or advanced green technology must achieve certification. 

Thus, the main objective of this study was to evaluate various extraction methods on the yield and properties of extracted carrageenans from a designated novel, haploid, female gametophyte of the *Kappaphycus alvarezii* strain G-N7. [15]. The comparisons will allow us to understand whether green technologies promote and maintain the expected properties and retain the necessary qualities of the certified food additive 

Carrageenan quality was measured using FTIR-ATR, coupled with viscosity and proximate chemistry (e.g., proteins, fatty acids, uronic acids, and carbohydrates) analyses. The goal was to determine which extraction method produced the best carrageenan yield and properties and which might be adapted for industrial-scale carrageenan production based on the Codex Alimentarius and EFSA regulations. Therefore, this study can provide insights to optimize production costs and recommend sustainable and environmentally friendly alternatives with enhanced efficiencies, as compared to traditional industrial methods.

## 2. Materials and Methods

### 2.1. Seaweed Cultivation and Harvest

The novel cultivar G-N7 is a haploid female gametophyte clonally propagated in an outdoor hatchery located in Bolinao, Pangasinan, Philippines. The specific life history phase, sex, and ploidy of G-N7 were established from the progeny of a wild, fertile diploid tetrasporophyte, known as KaTR-N, which was collected from Guiuan, Samar, Philippines [15]. Based on the cox2-3 spacer sequence, this strain belongs to KALV-3, a predominantly wild haplotype from the collection site. It is genetically distinct from the commercially cultivated haplotype 3, known commercially as the “Tambalang” strain, which is the cultivar mostly farmed for carrageenan processing [16] (Figure 1).

G-N7 was clonally propagated using branch cuttings in an aquarium (58 × 30 × 40 cm) supplied with air and flow-through, nutrient-replete, sand-filtered seawater pumped from a depth of approximately 12–15 m in the Guiguiwanen Channel, near the land-based hatchery. The outdoor hatchery, covered by translucent roofing, received semi-natural solar radiation, with an average noontime surface irradiance of 506 µmol photons m^−2^ s^−1^, measured using a cosine sensor coupled to a LI-COR light meter (LI-14000, LiCOR, Lincoln, NE, USA). The mean daytime water temperature was 31.7 °C, measured using HOBO data loggers (UA-002-64, HOBO, Onset Computer Corporation, Bourne, MA, USA) [17]. The average daily growth rate (DGR) in the nursery was approximately 2% per day. Biomass was harvested every 45–50 days or when the aquarium was filled with biomass. Harvested samples were sun-dried to an approximate 10% moisture content and packed in airtight plastic bags until transport for further chemical analyses.

### 2.2. Sample Preparation

The sun-dried, or commercially known as raw dried seaweed (RDS), biomass of the G-N7 *K. alvarezii* cultivar was, on receipt in the Portuguese laboratories, washed with distilled water to remove excess salt. The washed biomass (318 g) was dried in a forced air oven (Raypa DAF-135, R. Espinar S.L., Barcelona, Spain) at 60 °C for 48 h until its moisture content was <2%.

### 2.3. Carrageenan Extraction and Recovery

Carrageenan extraction was performed in triplicate using 1 g RDS per replicate. The extraction methods used were native or conventional, or ultrasound-assisted extraction (UAE) or supercritical fluid extraction (SFE) was applied, as described below. After extraction, processing for refined carrageenan (RC) and SRC adopted the following general protocols for precipitation, drying, and grinding for further analyses. Briefly, the carrageenan extract was precipitated by adding twice its volume of 96% ethanol and stirred. The precipitate was collected and drained using a cloth filter (mesh 60). This was washed and stored in 96% ethanol, at 4 °C for 48 h. Samples were then placed into Petri dishes and dried in a forced air oven (Raypa DAF-135, R. Espinar S.L., Barcelona, Spain) at 60 °C for 48 h, as described and determined by Pereira et al. [4,9]. Finally, dried carrageenan was weighed to determine the extraction yield (% of dry weight) and milled into powder (particle size < 0.05 cm) with a commercial grinder (TitanMill 300 DuoClean, Cecotec, Valencia, Spain) for further analyses.

### 2.4. Native Extraction (NE)

“Native” phycocolloid was extracted by placing 1 g of dry seaweed biomass (*n* = 3) in distilled water (100 mL), pH 7, at 100 °C for 3 h [4]. The solution was hot-filtered under vacuum through a cloth filter supported in a Buchner funnel, followed by a Gooch 2 silica funnel filtration. The extract was evaporated (rotary evaporator: 2600000, Witeg, Germany) under vacuum to one-third of the initial volume. This was followed by ethanol precipitation, drying, and grinding protocols to obtain carrageenan powder for further chemical analyses, as described below. 

### 2.5. Carrageenan Extraction via Refined and Semi-Refined Processes

In conventional (CE) and ultrasound-assisted (UAE) carrageenan extractions, (1) refined and (2) semi-refined processes were performed to obtain refined carrageenan (RC) and semi-refined carrageenan (SRC), respectively.

(1) Refining process—The solution was hot-filtered under vacuum through a cloth filter supported in a Buchner funnel. After this, the extract was filtered under vacuum with a Gooch 2 silica funnel. The extract was evaporated under vacuum to one-third of the initial volume (50 mL). Carrageenan was precipitated by adding twice its volume of 96% ethanol (100 mL). Due to gelling, after precipitation, carrageenan was collected through a filtration cloth. After this, the carrageenan fiber retained in the cloth was washed and stored with 96% ethanol for 48 h at 4 °C. Drying was conducted at 60 °C for 48 h, and fibers were milled into powder (particle size < 0.05 cm) using a commercial grinder (TitanMill 300 DuoClean, Cecotec, Valencia, Spain). This process was repeated three times to obtain triplicate replicates.

(2) Semi-refining process—The solution was hot-filtered under vacuum through a cloth filter supported in a Buchner funnel. The residue, retained in the cloth, was rinsed several times to remove the alkali and anything else that might dissolve in the water during the rinsing process. The alkali-treated seaweed was washed with 96% ethanol for 48 h at 4 °C. Drying, and grinding protocols were performed. The process was repeated three times to obtain triplicate replicates. 

#### 2.5.1. Conventional Extraction (CE)

##### 2.5.1.1. RC Obtention

Dry seaweed biomass (1 g, *n* = 3) was placed in a solution (150 mL) of NaOH (1 M) at 90–100 °C for 3 h, which was stirred every 15 min, according to the method described by Pereira and Van De Velde [18] (Figure 2). 

##### 2.5.1.2. SRC Sample Production

Dry seaweed biomass (1 g, *n* = 3) was placed in a solution (150 mL) of KOH (8%) at 75–80 °C for 2 h, which was stirred every 15 min, as shown in Figure 2. 

#### 2.5.2. Ultrasound-Assisted Extraction (UAE) Method

Extraction conditions were based on the previous method of Youssouf et al. [19] with modifications. Briefly, water was added to a heated pulsed ultrasound bath (120 W) (ultrasonic cleaner ULTR-3L2-001, IBX instruments, Barcelona, Spain) until the extraction beaker was in full contact with the external medium. After, temperature and time were set, before putting the extraction beaker (with seaweed and extraction solvent (NaOH or KOH)) in the ultrasound unit. After, the conventional pulsed ultrasound technique was applied to the extraction beaker with the following modified methodology (Section 2.5.2.1 and Section 2.5.2.2). 

##### 2.5.2.1. RC Sample Production

Dry seaweed biomass (1 g; *n* = 3) was placed in a solution (150 mL) of NaOH (1 M) and subject to a heated pulsed ultrasound bath (120 W) (ultrasonic cleaner ULTR-3L2-001, IBX instruments, Barcelona, Spain) at 70 °C for 60 min (Figure 3). 

##### 2.5.2.2. SRC Sample Production

Dry seaweed biomass (1 g; *n* = 3) was placed in a solution (150 mL) of KOH (8%) and subjected to ultrasound at different temperatures (45 °C and 70 °C) for 60 min (Figure 3). 

#### 2.5.3. Supercritical Water Extraction (SFE)

Supercritical water extraction was performed by placing forced air-dried seaweed (1 g, *n* = 3) and 100 mL distilled water into an electric pressure cooker (Aigostar 300008IAU, Aigostar, Madrid, Spain) at a temperature of 115 °C with an air pressure of 80 KPa for 2 h. The solution was hot-filtered under vacuum through a cloth filter supported in a Buchner funnel. Alcohol precipitation, drying, and grinding protocols were performed to obtain powdered RC for further analyses. 

### 2.6. Analyses of Extracted Carrageenans

#### 2.6.1. Yield 

The formula used to determine the carrageenan yield of each extraction is as follows:$$Yield=\frac{We}{Wds}\times 100$$
where *We* is the extracted carrageenan weight (g), and *Wds* is the dried seaweed weight (g) used for extraction.

#### 2.6.2. Fatty Acid Content

Carrageenan obtained from each extraction method was processed to obtain lipid extracts. Lipid extraction was performed in triplicate following the technique described by Gonçalves et al. [20]. Samples were incubated with methanol for the methylation of lipids. Then, n-hexane was added, and the samples were centrifuged to extract fatty acid methyl esters (FAMEs). The internal standard nonadecanoic acid C19 was added to each sample to quantify the fatty acid (FA) content and stored at −80 °C. FAME separation was carried out through gas chromatography‒mass spectrometry (GC‒MS) equipped with a 0.32 mm internal diameter, 0.25 μm film thickness, and 30 m long TR-FFAP column. The sample (1.00 μL) was injected into spitless mode. The initial column temperature was programmed at 80 °C and held for 3 min; the first ramp (20 °C min^−1^) increased the temperature to 160 °C. The second ramp (2 °C min^−1^) reached up to 190 °C, and the last ramp (5 °C min^−1^) reached a final temperature of 220 °C that was held for 10 min, as described by Gonçalves et al. [20]. Helium was used as the carrier gas, at a flow rate of 1.4 mL min^−1^. The identification of each peak was performed by comparing the retention time and mass spectrum of each FAME to those of the Supelco^®^37 component FAME mix (Sigma-Aldrich, Steinheim, Germany). The integration of the FAME peaks were carried out using the equipment’s software. The quantification of the FAMEs was performed as previously described in Gonçalves et al. [20].

#### 2.6.3. Carbohydrate and Uronic Acid Content

Carrageenan obtained from each triple replicated extraction method underwent hydrolysis in triplicate according to Selvendran et al. [21] to extract neutral sugars and uronic acids. These sugars were then reduced and acetylated as described by Coimbra et al. [22] to produce alditol acetates, which were subsequently analyzed by gas chromatography. The analysis was performed using a Thermo Scientific Trace 1310 chromatograph (Waltham, MA, USA) equipped with a flame ionization detector (GC-FID). A TG-WAXMS A GC column (30 m length, 0.32 mm i.d., 0.25 μm film thickness) was employed. The initial temperature of the oven was set at 180 °C and maintained for 1 min, then increased at a rate of 15 °C/min to 220 °C, held for 5 min, followed by a final ramp of 1 °C/min to reach 230 °C, where it was held for another 5 min. Helium served as the carrier gas at a flow rate of 1.7 mL/min. Peaks were identified by their retention times and quantified by comparison with standards. Uronic acids were measured using the Blumenkrantz and Asboe-Hansen method [23] with a Biochrom EZ Read 2000 Microplate reader (Biochrom Ltd., Cambridge, UK) at an absorbance wavelength of 520 nm. Galacturonic acid (Merck KGaA, Darmstadt, Germany) was used for the calibration curve, and 3-phenylphenol was the colorimetric reagent. The results are presented as the percentage of the dried polymer biomass.

#### 2.6.4. Protein Content

Carrageenan solutions (1% *m*/*v*) (*n* = 3) were prepared from powder obtained from the different extraction methods applied by dissolving the polysaccharide in distilled water. Protein content was measured after Bradford [24], based on spectrophotometry, adapted to the microplate. Samples were analyzed at 595 nm using a Biochrom EZ Read 2000 Microplate reader (Biochrom Ltd., Cambridge, UK). Protein concentration was calculated by comparison using bovine serum albumin (Merck KGaA, Darmstadt, Germany) as a standard. 

#### 2.6.5. Viscosity, pH, EC, and TDS

Carrageenan solutions (1% *m*/*v*; *n* = 1 from each extraction method) were prepared by dissolving the powdered polysaccharide in distilled water using a heating plate and magnetic stirring. Then, the solutions were cooled until reaching room temperature, and viscosity measurement was carried out using spindles SP2 and SP3 in an IKA Rotavisc Viscometer (KA-Werke GmbH & Co. KG, 79219 Staufen, Germany), with a speed of 100 rpm for 1 min. A pH/Conductivity/TDS meter (Combo HI98129, HANNA instruments, Smithfield, RI, USA) was used to measure the pH and TDS values of the carrageenan solutions. Commercial standards of kappa- and iota-carrageenans were obtained from Thermo Fisher Scientific (Waltham, MA, USA).

#### 2.6.6. Spectrophotometric Profiles of Carrageenan Solutions

Carrageenan solutions (1% *m*/*v*; *n* = 1) prepared for the viscosity analyses were diluted with distilled water (1:2), and UV-VIS absorption spectra were measured in the range of 200–800 nm using a UV-3100PC, UV/VIS Scanning Spectrophotometer (VWR^®^ Radnor, PA, USA) with 1 cm quartz cuvettes.

Attenuated total reflectance (ATR) Fourier transform infrared (FT-IR) spectroscopy was employed to characterize the structure of dried extracted carrageenans. The IR spectra (24 scans) were obtained at room temperature (referenced against air) in the wave number range of 400–4000 cm^−1^ (resolution of 4 cm^−1^) using a Bruker Alpha II (Bruker, Ettlingen, Germany). Spectra were analyzed with OPUS 7.2 software (Bruker, Ettlingen, Germany). Commercial standards of kappa- and iota-carrageenans were obtained from Thermo Fisher Scientific (Waltham, MA, USA). The ratios between the 805 and 845 cm^−1^ absorption bands in the spectra were calculated [4] and used to determine the degree of iota/kappa hybridization [4].

### 2.7. Statistical Analyses

All experiments and subsequent analyses were performed in triplicate, and data are presented as the means ± standard deviations. An analysis of variance (ANOVA) was performed, and comparisons of means were conducted using different multiple comparison tests using the Sigmaplot program (version 14.0, SigmaPlot, D-40212 Düsseldorf, Germany). Values were considered to differ significantly if the *p* value was <0.05.

For the exploratory analysis, a multi-linear regression (singular value decomposition) was used to estimate the relationship between each sample and the standard commercial kappa-carrageenan and iota-carrageenan using the Spectragryph program (Spectragryph, 87561 Oberstdorf, Germany) [25] in the spectral range of 400–4000 cm^−1^.

## 3. Results

### 3.1. Extraction Yield and Biochemical Composition

The mean yield of carrageenans obtained using different protocols ranged from 33.73 ± 10.52 to 77.33% of the initial dry weight of the sample (detailed in Table 1). The highest mean yield (i.e., 77.3 ± 2.5%) was achieved using the alkali (KOH)-treated conventional method, while the lowest yield (33.73 ± 10.52%) came from the alkali (NaOH)-treated UAE method. Despite the 44% difference between the minimum and maximum mean yields, the variations among the different extraction protocols were not statistically significant, though they could have economic implications.

In general, the yield under an alkaline (KOH) environment was higher for both the conventional (CE; 90 °C, 2 h) and ultrasound-assisted (UAE; 120 W, 1 h) extraction methods. UAE (120 W, 1 h) revealed values close to the conventional carrageenan yield in a shorter time.

The protein content (Table 1) in all treatments showed very low values, with UAE (NaOH) exhibiting the highest content (0.04 ± 0.02%), which is very important for analyzing extraction quality and obtaining the approval of carrageenan as a human food additive ingredient.

Uronic acid content analysis (Table 1) revealed significant differences amongst the extraction methods. The NE, SFE, and CE (NaOH) methods showed a notably higher mean uronic acid content (i.e., 13.59 ± 1.97, 13.52 ± 1.37, and 13.95 ± 1.05% of DW, respectively), as compared to the others. In contrast, the UAE (NaOH) method resulted in the lowest uronic acid content.

A higher total content of FAMEs was observed in KOH alkali-treated RC (Table 1). FAME analysis identified two saturated fatty acids (SFAs) (palmitic acid [C16:0] and stearic acid [C18:0]) and one monounsaturated fatty acid (MUFA) (oleic acid [C18:1]). While palmitic acid (C16:0) values did not show statistically significant differences, all KOH alkali treatments exhibited higher mean values. Stearic acid (C18:0) values were very similar across extraction methods, and oleic acid (C18:1) was detected only in the conventional extraction method [CE (NaOH) and CE (KOH)] and UAE (KOH45). 

The total monosaccharide content (Table 1) was higher for the alkali treatments performed with KOH [CE (KOH), UAE (KOH45), and UAE (KOH)]. Carbohydrate analysis identified four different monosaccharides in addition to galactose: glucose, fucose, arabinose, and xylose. Galactose was the most abundant residue in all treatments. Glucose had the second highest value, detected in all KOH alkali extractions [CE (KOH), UAE (KOH45), and UAE (KOH)] and in very low amounts in UAE (NaOH.

### 3.2. Viscosity, pH, EC, and TDS

The physicochemical parameters measured are summarized in Table 2. All 1% solutions of carrageenan extracts obtained through alkali (UAE KOH) treatment exhibited non-dissolvable particles and had a yellowish appearance, except for the commercial samples, which were clear. Solutions made with commercial iota- and kappa-carrageenan had the lowest EC and TDS values. Carrageenan extracted using the alkali (UAE; KOH; 70 °C) method had the highest viscosity at 658.7 cP. The pH of all samples from different extraction methods was alkaline, except for SFE, which had a slightly acidic pH (6.79), and NE, which had a neutral pH (7.34). The EC and TDS of alkaline (UAE; NaOH)-treated extracts were higher compared to the others, with values in a similar range.

### 3.3. UV‒VIS Absorption Spectra of Carrageenan Solutions

The carrageenans obtained through the different methods differed in their spectral profiles, suggesting that the composition and the UV absorbance potential varied depending on the extraction method (Figure 4).

Only the native (NE) carrageenan solution absorbed part of UV-A radiation (320–400 nm), exhibiting a peak at 321 nm.

The SFE, UAE (NaOH), UAE (KOH), UAE (NaOH), and UAE (KOH45) extracts showed peaks at 202, 204, 206, and 206 nm, respectively.

No pigments were detected in the carrageenan solutions.

### 3.4. FTIR-ATR Analysis

The FTIR-ATR spectra of commercial iota-carrageenan and kappa-carrageenan (Figure 5a and Figure 5b, respectively) were used as baselines for a comparison with the spectral profiles of carrageenans extracted using different methods (Figure 5c–i). The FTIR-ATR band assignments, letter code nomenclature, and band identification of the extracted carrageenans are detailed in Table 3.

The broad band around 1220 cm^−1^ (between 1210 and 1260 cm^−1^) is characteristic of sulfate esters in general and serves as a good indicator of total sulfates (sulfate esters) in sulfated polysaccharides [9]. This band is strong in the carrageenan standards, as shown in the spectra (Figure 5a,b) [26]. Several vibrational bands are characteristic of carrageenans. The absorption band at 930 cm^−1^ indicated the presence of 3,6-anhydrogalactose (DA), which is seen in the spectra of kappa- and iota-carrageenans. Additionally, the band at 845 cm^−1^ is related to the presence of D-galactose-4-sulfate (G4S), characteristic of kappa-, mu-, iota-, and nu-carrageenans. Conversely, the band at 805 cm^−1^, associated with the sulfate ester at position 2 of anhydrous-D-galactose (DA2S) residues, is only observed in the spectra of iota-carrageenan.

The weak bands in the 770 cm^−1^ region are related, according to Matsuhiro [27], to the skeleton bending of pyranose present in the carrageenan structure. Additionally, the band at 1150 cm^−1^ may be assigned to the C-O and C-C stretching vibrations of the pyranose ring common to all polysaccharides [18].

Going further on the spectra, the band at 1639 cm^−1^ is known to be an indicator of amide I, the presence of H_2_O, or proteins CO-NH/amide II from proteins [28].

The FTIR-ATR spectra of extracted carrageenans from the novel haploid *K. alvarezii* strain using various methods present absorption bands at the 930 cm^−1^ region and at the 845 cm^−1^ region, which were both considered typical and revealed the presence of kappa-carrageenan.

Conventionally extracted [CE (NaOH)] carrageenan (Figure 5f) showed a slightly higher shoulder at 930 cm^−1^ in the spectra and a decrease in the ratio 805/845, as compared to native carrageenans, was verified, meaning that there was a decrease in the iota fraction relative to the kappa fraction (Table 4).

The FTIR-ATR spectra of conventionally extracted [CE (KOH)] carrageenans (Figure 5c) were different from those of CE (NaOH), showing a slightly more visible band in the 805 cm^−1^ region (DA2S). An increase in the iota/kappa ratio was verified in comparison to native carrageenan and overall alkali extracted (NaOH) carrageenans, which corresponded to an increase in the iota fraction relative to the kappa fraction.

Ultrasound-assisted extracted refined carrageenans [UAE (NaOH)] (Figure 5g) were similar to those conventionally extracted [CE (NaOH)], and although the peak at 845 cm^−1^ was less sharp, the ratio 805/845 remained the same (Table 4).

Ultrasound-assisted extracted semi-refined carrageenans [UAE (KOH)] and UAE (KOH45) (Figure 5e and Figure 5d, respectively) revealed very similar spectra, with UAE (KOH) presenting a slightly higher peak in the 845 cm^−1^ region, which was visible in the lower iota/kappa ration in comparison to UAE (KOH45) (Table 4).

Bands at the 970–975 cm^−1^ regions were related to galactose (G) and were present in all samples, with CE(NaOH), NE, and SFE (Figure 5f, Figure 5h, and Figure 5i, respectively) showing more absorption in comparison to other samples.

Comparing the ratio of iota/kappa (Table 4) of each extraction method, with the iota-(0.82) and kappa-carrageenan (0.51) commercial samples, values ranged from 0.65 to 0.79.

Table 5 presents the absolute spectral similarities between carrageenans extracted from the various methods and each commercial sample. Commercial kappa-carrageenan samples revealed higher absolute spectral similarity than iota-carrageenan in all carrageenan samples obtained through the different extraction methods. However, iota-carrageenan still showed relatively high values. Carrageenan extracted through CE (KOH) showed the highest mean value of 64.66%.

## 4. Discussion 

The yield of native carrageenan obtained was slightly higher compared to that in most published studies using different cultivars (Table 6). The content and quality of polysaccharides in seaweeds can vary significantly with the season and are influenced by various biotic and abiotic factors [29,30]. Although there are many reports on carrageenan yields from *K. alvarezii*, making quantitative and qualitative comparisons is challenging due to variations in extraction methods and the strains or cultivars of the species. When comparable extraction conditions are used, the observed higher yield may be attributed to the higher carrageenan content of the specific strain. The lack of statistically significant differences demonstrates that the seaweed biomass had high heterogeneity, due to the mean range from 34 to 77, although the methods demonstrated significant carrageenan extraction power (Table 6) when compared to other studies, as presented in Table 6.

The CE (NaOH) method presented close but lower values (35.67 ± 1.89% dw of carrageenan extracted) than previously reported (48%), while CE (KOH) had higher values (77.33 ± 2.49% dw of carrageenan extracted). It is important to note that this higher yield measured in the CE (KOH) treatment could be because residual cellulose from the cell walls remained in the carrageenan fibers after washing, which only eliminates residual minerals, proteins, and lipids. SRC production consists of a method where carrageenan is never extracted from seaweed, but non-polysaccharide compounds are cleaned from seaweed. Thus, other polysaccharides (such as cellulose) are present in SRC.

According to a study conducted by Rafiquzzaman et al. [14] on carrageenan extracted from *Hypnea musciformis*, the yield was higher using the aqueous UAE (500 W, 20 min) method, as compared to the conventional aqueous extraction method (native extraction). The present work compared conventional methods with ultrasound-assisted extraction, CE (NaOH) (35.67 ± 1.89% dw of carrageenan extracted) vs. UAE (NaOH) (33.73 ± 10.52% dw of carrageenan extracted), and CE (KOH) (77.33 ± 2.49% dw of carrageenan extracted) vs. UAE (KOH) (76.70 ± 1.44% dw of carrageenan extracted). The ultrasound-assisted method gave slightly lower extraction yields than conventional extraction. However, these were performed in a shorter time (i.e., one-third of the time in NaOH solution and half of KOH) and at a lower temperature. Such results can be attributed to the effective disruption of cell walls, reduction in particle size, and increased mass transfer of cell contents [32]. Looking at the two performed ultrasound-assisted extraction methods using KOH, UAE (KOH45) (63.20 ± 3.23), the yield was lower than UAE (KOH) (76.70 ± 1.44). Conversely, it operated at almost half of the extraction temperature (45 °C). A less abrupt reduction in the yield may be due to a lower level of possible degradation of the polysaccharide due to temperature.

The mean SFE carrageenan value (53.40 ± 1.80% dw of carrageenan extracted) was lower than that reported by Gereniu et al. [12] but still higher than CE (NaOH) and UAE (NaOH).

The biochemical analyses performed on the extracted carrageenans provided information on both composition and quality. There are several studies on the biochemical composition of the dried biomass of different *K. alvarezii* strains. However, there are very few studies on the polysaccharide of this species, specifically the FAME content. The FAME composition and content of carrageenans extracted by the different methods in this study were used to understand to what extent the extraction method was able to remove these compounds. A total of three fatty acids were detected: two saturated fatty acids (SFAs) and one monounsaturated fatty acid (MUFA). These results, as expected, were lower than those reported for the raw biomass of *K. alvarezii* in previous studies but present the same proportions as previous results [12,34], with saturated fatty acids (SFAs) presenting higher contents, especially palmitic acid (C16:0) and stearic acid (C18:0), followed by monounsaturated fatty acid (MUFA) oleic acid (C18:1). From all extraction methods, the CE (NaOH) extract produced the highest fatty acid content, followed by that of UAE (KOH), UAE (KOH45), and NE. These values were even lower, if before extraction, the ground dry material was pre-treated with a mixture of solvents (such as acetone and methanol). In the present work, this was not applied, and the values are still acceptable for human food approval.

The present work reports on the carbohydrate composition of carrageenans from a novel *Kappaphycus alvarezii* strain (G-N7). The obtained results were compared to those of previous studies, such as Rhein-Knudsen et al. [35], which evaluated the monosaccharide composition of native and alkali-extracted carrageenan of *K. alvarezii* from Nha Trang, Vietnam, by HPAEC-PAD analysis. The values obtained in their study presented high contents of galactose (68–70% carrageenan dw) and a low content of glucose (3–6% carrageenan dw), and other monosaccharides were detected at vestigial contents (2% carrageenan dw), namely, fucose, xylose, and mannose. In another study conducted by Meinita et al. [36], the total carbohydrate content of raw seaweed *K. alvarezii* was obtained from various places in Indonesia, and the mean values were even higher, ranging from 35 to 78%, by weight. These variations seem to be related to the type of extraction method performed, geographic origin, strain/cultivar, and harvest time [30].

No previous studies determined the uronic acid content in carrageenan extracted from *K. alvarezii*, but in comparison to analyses performed on the dried seaweed biomass of other red seaweed species, namely *Asparagospis armata*, *Calliblepharis jubata*, *Chondracanthus teedei* var. *lusitanicus*, and *Grateloupia turuturu*, the values reported were very low [37]. For the presence of uronic acids, in FTIR-ATR-obtained spectra, a clear peak around 1700 cm^−1^ would be expected, but this was not the case. Nevertheless, this did not necessarily mean that the colorimetric reaction was not effective. Depending on the seaweed, the carboxyl groups of uronic acids can react with carbohydrates (which can reduce sulfate esters in polysaccharides), as previously reported in the red seaweed *Hypnea musciformis* [38] and described in pattern detection [39].

In general, the alkali extractions performed with KOH showed lower values of galactose, possibly due to the benefit to 3,6-anhydrogalactose. This corresponded to the conversion of the 4-linked galactose-6-sulfate in native samples to anhydro-galactose in the alkali-extracted carrageenans. Thus, the biological precursors mu- and nu-carrageenan were converted into the kappa- and iota-carrageenan forms, respectively [4].

The protein content of all samples presented very low values, similar to commercial carrageenans [40], meeting the Food and Agriculture Organization of the United Nations (FAO) quality criteria, which require values lower than 2%.

Regarding the measured physicochemical parameters, in addition to protein content, carrageenan pH and viscosity were also monitored and regulated parameters used to assess the quality of commercially produced polysaccharides, and the value must be greater than 5 cP to meet FAO quality criteria. The results from the present work showed that, relative to pH criteria, the extracted carrageenan from the NE, SFE, and alkali (NaOH) methods were not suitable for food applications due to having a pH < 8 and thus cannot be adopted for an industrial-scale extraction process as the resulting product does not conform with the food additive regulations. 

The viscosity varied greatly (from 7.8 to 658.7 cP). This could be due to the different independent variables, e.g., (1) extraction time, (2) alkali treatment, (3) type of alkali (NaOH or KOH), (4) extraction method (NE, UAE, or SFE), and (5) temperature.

SFE showed the lowest viscosity value (7.8 cP), possibly due to irregularities in the chain, as reported by BeMiller [41], caused by the high temperature and pressure, causing carrageenan to lose its properties. Higher viscosity values are usually attributed to carrageenans obtained by extraction methods using an alkali solution of KOH. Bono et al. [42] investigated the influence of process conditions on the viscosity of conventional alkali KOH-extracted carrageenan from *K. alvarezii* and determined the optimal conditions to be 80 °C for 30 min and a 10% *w*/*w* KOH solution, which resulted in a gel viscosity of 1291.84 cP. In the present work, UAE (KOH) revealed the highest value of viscosity (658.7 cP at 1%). Comparing UAE (KOH) with UAE (KOH45), these higher values might be due to temperature, since it is the only parameter varying between them. The high viscosity value can also mean that higher sulfate levels are present. According to Bono et al. [42], the viscosity of carrageenans can be influenced by the levels of, and directly proportional to, sulfate. A higher sulfate content resulted in a higher viscosity, this being due to the ability of the sulfate group to exert a repulsion force between negative charges along the polymer chain, and as a result, the molecular chain stiffens, and hence, the viscosity increases [43]. The neutral-pH and low-viscosity carrageenan obtained using SFE suggest that it has limited applications for the food industry, due to not complying with the carrageenan safety and quality control check (human food-approved carrageenan needs to have a pH above 8 and a, at least, viscosity above 10 cP). However, this neutral and low-viscosity carrageenan can be useful for exploring its biostimulant properties in agriculture. 

No studies regarding EC and TDS values from carrageenan solutions of *K. alvarezii* were found. Nevertheless, in comparison to the commercial carrageenans used as standards, all values were higher, especially in UAE (NaOH), CE (NaOH), CE (KOH), and UAE (KOH45), possibly indicating high levels of charged ions and substances not soluble or precipitated.

Among the UV‒VIS spectra of different carrageenan solutions, only the native extracted carrageenan samples absorbed part of the UV-A radiation (320–400 nm). As previously reported, red seaweeds accumulate photoprotective compounds with ultraviolet radiation absorption capabilities, such as mycosporine-like amino acids (MAAs), which absorb in this specific UV region [44]. The UV absorption spectrum with prominent peaks between 320 and 340 nm is in accordance with the presence of MAAs absorbing in this range [44]. This finding revealed a possible application of the novel strain of *K. alvarezii* in para-pharmaceutical and cosmetic industries for UV protection. The NE, CE (KOH), UAE (KOH; 45 °C), and UAE (NaOH) extracts showed a peak between 200 and 270 nm, which corresponds to polysaccharides bound covalently with aromatic compounds [45]. Additionally, these results were partially correlated with the protein content. The lack of pigment detected in the UV‒VIS spectra supported the purity of the extracted carrageenans and reinforced that a de-pigmentation step in the extraction processes of this seaweed strain was not required.

The FTIR-ATR spectra of extracted carrageenan from the *K. alvarezii* (G-N7) strain using various methods revealed the presence of kappa- and iota-carrageenans, especially in alkali-extracted CE (NaOH) and UAE (NaOH) (Figure 5h, Figure 5i, Figure 5f, and Figure 5g, respectively), indicating that the novel strain presents a hybrid kappa/iota-carrageenan, as verified in previous studies in several other *K. alvarezii* strains [18]. This is easily understood, especially in the iota/kappa ratio (Table 5), with CE (NaOH) and UAE (NaOH) presenting a lower value (0.65), meaning a lower content in iota-carrageenan, while UAE (KOH45) presented the highest ratio (0.79), and there was a higher content in iota-carrageenan. Bands in the 970–975 cm^−1^ region were identified as galactose (G), especially in CE (NaOH), NE, and SFE (Figure 5f, Figure 5h, and Figure 5i, respectively), and showed more absorption in comparison to the rest of the samples, which corroborated the results obtained in its quantification.

Similarity values provided valuable insights regarding carrageenan composition. Nonetheless, it is important to consider that similarity is relative in all spectra. The fact that sample G-N7 contained hybrid carrageenan is highlighted, and from the higher values of the kappa fraction, we could conclude that there was a higher content; however, the similarity regarding all spectra could be due to other less determinant peaks, and iota-carrageenan still showed relatively high values. This study demonstrated that KOH extraction can reduce processing time and enhance industrial efficiency, and although KOH extraction is a semi-refined extraction method, it demonstrated good quality when compared to the standards set by the carrageenan regulatory body. The carrageenan viscosity properties (although measured only once here) can be very important for carrageenan applications in various types of drug delivery and tissue engineering. This study demonstrated that some extraction techniques altered not only the purity but also the physicochemical properties of extracted colloids. Therefore, the properties could perhaps be more easily altered during extraction. This would be conducted as opposed to chemical modifications after extraction but before integration in pharmaceutical, drug, and medical applications. 

## 5. Conclusions

This study is the first to evaluate the potential of the novel haploid *Kappaphycus alvarezii* G-N7 strain as a new seedstock for commercial cultivation and biomass yield for carrageenan production. Carrageenans extracted using conventional, UAE, and SFE methods were subjected to comprehensive physico-biochemical analyses, with values compared to benchmark, commercial samples of standard-grade kappa- and iota-carrageenans. UAE (KOH) extracts demonstrated a similar yield to those obtained from the conventional method but required less than half the extraction time and produced improved carrageenan viscosity. The extraction method (UAE (KOH)) presented here can reduce carrageenan extraction costs with an enhanced control of the chemical characterization of the polymer produced.

This study found kappa- and iota-carrageenans with comparable iota/kappa ratios and protein yields, particularly in alkali-extracted CE (NaOH) and UAE (NaOH). The samples yielded less carrageenan (CE (NaOH) (35.67 ± 1.89)) than UAE (NaOH) (33.73 ± 10.52). However, UAE (NaOH) showed variances in viscosity and conductivity (when compared to the other extraction methods), which might be troublesome in the food business, due to carrageenan physiochemical safety check regulations. Thus, UAE (NAOH) carrageenan can have different behavior and chemical hazards from carrageenan approved as a human food additive. This question can be answered by employing an osmosis procedure to lower the ion concentration in carrageenan. Thus, these novel methods which are not currently used by the phycocolloid industry need more research before being applied at the industrial scale, due to strict regulations required for carrageenan as a *(human) food additive ingredient.

The FTIR-ATR spectra of the extracted carrageenans matched the commercial benchmark samples of both standard-grade iota- and kappa-carrageenans, although the manufacturing method was not specified by the supplier (Thermo Scientific). 

Consequently, the clonally propagated novel strain of *Kappaphycus alvarezii* represents a promising raw material for future commercial carrageenan production for the food industry as an additive, with potential pharmacological and cosmetic applications.

In the future, there is a need to conduct more extensive chemical and physicochemical studies (with more replications), applying different techniques, such as HPLC, NIR, and XRF to analyze molecular weight, mineral content, and other compounds in the carrageenan extracted.

## Figures and Tables

**Figure 1 marinedrugs-22-00491-f001:**
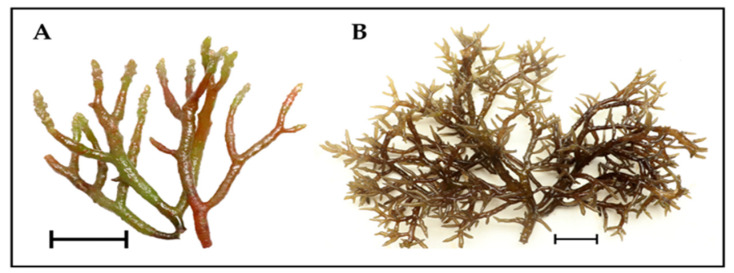
*Kappaphycus alvarezii* cultivars. (**A**) Novel haploid female gametophyte G-N7 strain. (**B**) Commercially cultivated “Tambalang” strain. Scale bar = 5 cm.

**Figure 2 marinedrugs-22-00491-f002:**
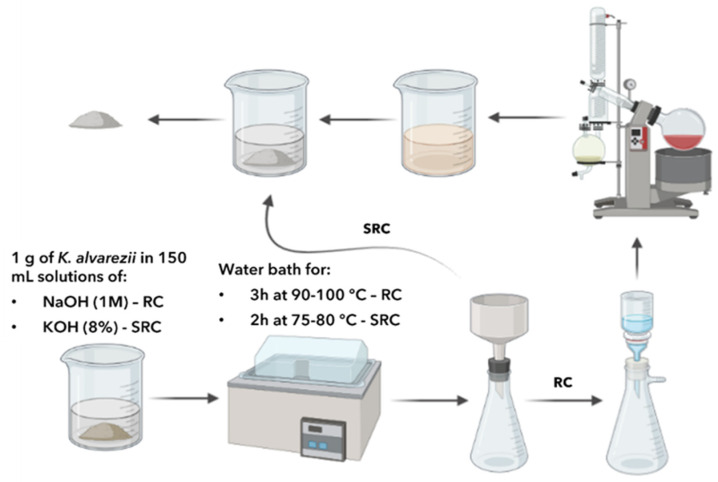
Workflow of conventional extraction method performed for carrageenan extraction (RC: refined extraction; SRC: semi-refined extraction).

**Figure 3 marinedrugs-22-00491-f003:**
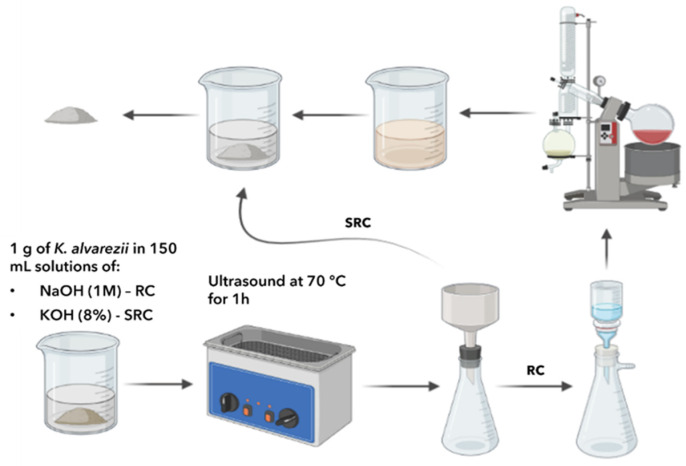
Workflow of ultrasound-assisted extraction method performed for carrageenan extraction (RC: refined extraction; SRC: semi-refined extraction).

**Figure 4 marinedrugs-22-00491-f004:**
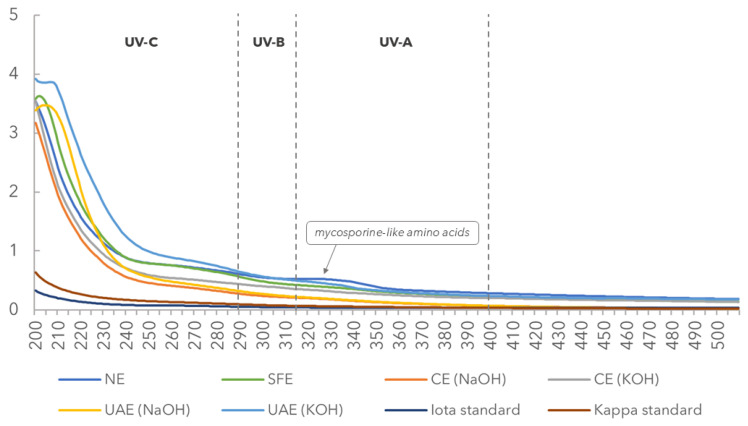
UV absorption spectra (λ = 200–600 nm) of carrageenan solutions extracted by native extraction (NE), conventional extraction (CE: NaOH and CE: KOH), ultrasound-assisted extraction (UAE: NaOH, 120 W, 70 °C, 1 h; UAE: KOH, 120 W, 45 °C, 1 h; UAE: KOH, 120 W, 70 °C, 1 h), and supercritical water extraction (SFE). UV-C range: 100–290 nm; UV-B range: 290–315 nm; and UVA range: 315–400 nm.

**Figure 5 marinedrugs-22-00491-f005:**
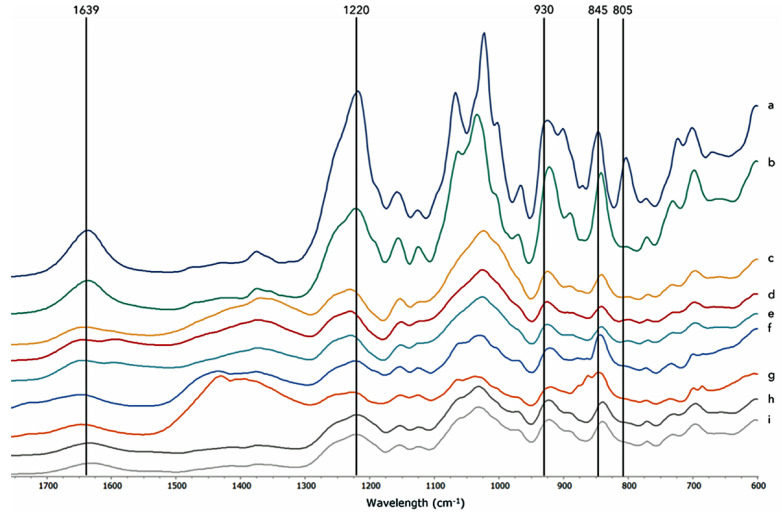
FTIR-ATR spectra of carrageenan standards: (**a**) iota-carrageenan and (**b**) kappa-carrageenan from Thermo Scientific and carrageenans from the novel strain of *Kappaphycus alvarezii* obtained through the various extraction techniques: (**c**) conventional extraction (KOH); (**d**) ultrasound-assisted extraction (UAE) (KOH 45); (**e**) ultrasound-assisted extraction (KOH); (**f**) conventional extraction (NaOH); (**g**) ultrasound-assisted extraction (NaOH); (**h**) native extraction; and (**i**) supercritical water extraction.

**Table 1 marinedrugs-22-00491-t001:** Carrageenan extraction methods versus yield, protein content, uronic acid content, and the relative and total composition of fatty acids and monosaccharides (wt%) present in carrageenan extracted using different methods. Data are given as the mean ± SDs. Values with the same letter are not significantly different (*p* > 0.05).

Extraction Method								
Parameters		NE	SFE	CE (NaOH)	CE (KOH)	UAE (NaOH)	UAE (KOH45)	UAE (KOH)
Extraction yield (%)		45.47 ± 1.92 ^a^	53.40 ± 1.80 ^a^	35.67 ± 1.89 ^a^	77.33 ± 2.49 ^a^	33.73 ± 10.52 ^a^	63.20 ± 3.23 ^a^	76.70 ± 1.44 ^a^
Protein (%)		0.00 ± 0.01 ^b^	0.01 ± 0.01 ^a,b^	0.02 ± 0.01 ^a,b^	0.01 ± 0.01 ^a,b^	0.04 ± 0.02 ^a^	0.02 ± 0.01 ^a,b^	0.01 ± 0.01 ^a,b^
Uronic acids (%)		13.59 ± 1.97 ^a^	13.52 ± 1.37 ^a^	13.95 ± 1.05 ^a^	9.43 ± 2.78 ^a,b^	6.43 ± 0.11 ^b^	8.95 ± 0.23 ^a,b^	10.06 ± 1.28 ^a,b^
Fatty acids (%)	C16:0	0.02 ± 0.01 ^a^	0.01 ± 0.00 ^a^	0.02 ± 0.01 ^a^	0.05 ± 0.02 ^a^	0.01 ± 0.00 ^a^	0.07 ± 0.02 ^a^	0.07 ± 0.00 ^a^
	C18:0	0.01 ± 0.01 ^a^	0.01 ± 0.00 ^a^	0.02 ± 0.00 ^a^	0.01 ± 0.00 ^a^	0.01 ± 0.00 ^a^	0.01 ± 0.00 ^a^	0.01 ± 0.00 ^a^
	C18:1	nd	nd	0.01 ± 0.01 ^a^	0.01 ± 0.00 ^a^	nd	0.01 ± 0.00 ^a^	nd
	Σ	0.03	0.03	0.05	0.07	0.03	0.09	0.08
Monosaccharides (%)	Galactose	6.98 ± 0.23 ^a,b^	7.28 ± 0.29 ^a,b^	7.87 ± 0.36 ^a^	5.23 ± 0.30 ^c^	6.45 ± 0.37 ^b^	5.06 ± 0.05 ^c^	6.29 ± 0.34 ^b^
	Glucose	nd	nd	nd	1.92 ± 0.08	0.04 ± 0.06	1.87 ± 0.19	2.20 ± 0.03
	Fucose	0.21 ± 0.03 ^b,c^	0.15 ± 0.02 ^c^	0.36 ± 0.09 ^b,c^	0.68 ± 0.02 ^a^	0.31 ± 0.12 ^b,c^	0.44 ± 0.11 ^a,b^	0.34 ± 0.11 ^b,c^
	Arabinose	nd	nd	nd	0.03 ± 0.01 ^a^	nd	0.06 ± 0.02 ^a^	0.08 ± 0.00 ^a^
	Xylose	0.10 ± 0.00 ^b,c^	0.05 ± 0.03 ^c^	0.24 ± 0.03 ^a^	0.32 ± 0.02 ^a^	0.23 ± 0.08 ^a^	0.19 ± 0.02 ^a,b^	0.05 ± 0.03 ^c^
	Σ	7.29	7.49	8.48	10.10	7.06	9.49	11.16

nd—non-detectable, value below the detection limit.

**Table 2 marinedrugs-22-00491-t002:** Physicochemical parameters of the *Kappaphycus alvarezii* haploid strain solution (1%): viscosity, pH, electrical conductivity (EC), and total dissolved solids (TDS).

Extraction Method	Viscosity (cP)	pH	EC (µS cm^−1^)	TDS (ppm)
NE	16.8	7.34	2373	1158
SFE	7.8	6.79	2323	1170
CE (NaOH)	15.9	10.25	3156	1604
CE (KOH)	50.87	10.85	2733	1366
UAE (NaOH)	8.1	10.70	>3999	>2000
UAE (KOH45)	183.6	10.89	2817	1439
UAE (KOH)	658.7	9.30	2492	1286
Iota standard (i)	134.7	9.69	2019	1012
Kappa standard (k)	79.2	8.66	1891	949

**Table 3 marinedrugs-22-00491-t003:** FTIR band assignment (wave number cm^−1^), letter code nomenclature, and band identification of carrageenans obtained through various methods.

Wave Number (cm^−1^)	Bound	Letter Code	Iota (ι)	Kappa (κ)	NE	SFE	CE (NaOH)	CE (KOH)	UAE (NaOH)	UAE (KOH45)	UAE (KOH)
1210–1260	Sulfate ester (S=O)	S	1219	1222	1220	1221	1223	1231	1227	1231	1230
928–933, 1070 (shoulder)	3,6-anhydro-D-galactose	DA	925.2 (1067)	922.3 (1063)	923.1	923	921.5	924.6	920.9 (1063)	925.7	925.1
970–975	Galactose	G/D	967	971.3	972.8	972.8	-	-	-	-	-
890–900	β-D-galactose-de-sulfated	G/D	902.1	889.9	-	-	-	890.9	-	-	888.5
840–850	D-galactose-4-sulfate	G4S	846.4	842.2	839.7	839.7	842.4	841.5	846.2	841.6	841.4
830	D-galactose-2-sulfate	G2S	-	-	-	-	-	-	-	-	-
820, 825 (shoulder)	D-galactose-2,6-disulphate	D2S, D6S	-	-	-	-	-	-	-	-	-
810–820, 867 (shoulder)	D-galactose-6-sulfate	G/D6S	-	-	-	-	-	-	-	-	-
800–805, 905 (shoulder)	3,6-anhydro-D-galactose-2-sulfate	DA2S	803.4	-	-	-	-	800.2	-	800.7	800.9

- not detected.

**Table 4 marinedrugs-22-00491-t004:** Iota/kappa ratios of carrageenans extracted by various methods and each commercial sample.

	Iota (ι)	Kappa (κ)	NE	SFE	CE (NaOH)	CE (KOH)	UAE (NaOH)	UAE (KOH45)	UAE (KOH)
iota/kappa ratio	0.82	0.51	0.68	0.66	0.65	0.73	0.65	0.79	0.78

**Table 5 marinedrugs-22-00491-t005:** Absolute spectral similarity between carrageenans extracted from various methods and each commercial sample.

	Similarity (%)						
	NE	SFE	CE (NaOH)	CE (KOH)	UAE (NaOH)	UAE (KOH45)	UAE (KOH)
Iota (ι)	33.59	32.17	47.39	53.83	38.82	43.18	41.55
Kappa (κ)	40.10	38.35	56.83	64.66	46.92	51.98	49.93

**Table 6 marinedrugs-22-00491-t006:** Extraction yield mean (%) of carrageenan extracted from *Kappaphycus alvarezii* of different localities using different methods.

Extraction Conditions	Yield Mean (%)	Locality	Reference
Aqueous (Native)	40–50	Cam Ranh Bay, Vietnam	[31]
Aqueous (SFE)	71	Wagina, Solomon Islands	[11]
Alkali (NaOH)	48	Palk Bay, India	[32]
Alkali (KOH)	52	Palk Bay, India	[32]
Alkali (4% KOH)	53.2	Philippines (commercial sample)	[33]
Alkali (6% KOH)	54.6	Philippines (commercial sample)	[33]
Alkali (8% KOH)	53.7	Philippines (commercial sample)	[33]
Aqueous UAE	50-55	BIS Algoculture (Madagascar)	[19]
Various (7 methods)	34–77	Philippines (novel variant)	This study, Table 1

## Data Availability

Data can be made available by the author on request.

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
