# Peer review of "Advanced Extraction Techniques and Physicochemical Properties of Carrageenan from a Novel Kappaphycus alvarezii Cultivar"

_marinedrugs, 2024, doi:10.3390/md22110491_

Round 1

Reviewer 1 Report

Comments and Suggestions for Authors

The novelty and significance of this work should be made more explicit, especially in the abstract and conclusion sections. There is no sufficient discussion of the results obtained. After minor revisions, I recommended publishing this work in the Marine Drugs.

Detailed remarks about the text are as follows:

The critical data obtained from the study should be given in the abstract section. The data obtained in the abstract section are mentioned very superficially.

The introduction section is too long. Unnecessary parts should be removed by focusing on topics related to the study's content.

2.3. Carrageenan extraction and recovery: Extraction is the critical part of this study. No information has been provided in this regard. How were conditions determined? Was it determined in the researchers' previous work? If so, please indicate references; otherwise, discuss how these conditions were chosen, which is very important for the results.

Figure 2: This figure should either be more detailed (temperature, duration, etc.) or deleted. It is not very meaningful in this form.

The Results and Discussion section contains many tables, but the results lack in-depth discussion. Explain the trends observed in your results within this section. Provide more interpretation.

Author Response

Reviewer 1

The novelty and significance of this work should be made more explicit, especially in the abstract and conclusion sections. There is no sufficient discussion of the results obtained. After minor revisions, I recommended publishing this work in the Marine Drugs.

Detailed remarks about the text are as follows:

  1. The critical data obtained from the study should be given in the abstract section. The data obtained in the abstract section are mentioned very superficially.

Answer: It was revised

  1. The introduction section is too long. Unnecessary parts should be removed by focusing on topics related to the study's content.

Answer: It was revised to objectives of the study.

  1. 3. Carrageenan extraction and recovery: Extraction is the critical part of this study. No information has been provided in this regard. How were conditions determined? Was it determined in the researchers' previous work? If so, please indicate references; otherwise, discuss how these conditions were chosen, which is very important for the results.

Answer: This was addressed in the revised methods.

  1. Figure 2: This figure should either be more detailed (temperature, duration, etc.) or deleted. It is not very meaningful in this form.

Answer: The figure was deleted

  1. The Results and Discussion section contains many tables, but the results lack in-depth discussion. Explain the trends observed in your results within this section. Provide more interpretation.

Answer: The reviewer’s comments were incorporated in the revised manuscript.

Reviewer 2 Report

Comments and Suggestions for Authors

I send enclosed the comments for authors.

Comments on the Quality of English Language

Check spelling mistakes (units).

Author Response

Reviewer 2

General comments:

The paper has potential but must be thoroughly revised and corrected.

The text should be revised and rewritten for a better understanding of the main objective of this research and the benefits of this knowledge for society.

References should be 40 -45 at most and be related to the text.

Check units and statistical analysis.

Answer: We made a full revision of the manuscript according to the reviewer’s comments and we re-checked all the units validated the statistical analyses.

Specific comments:

Introduction:  

Line 49: Reference [2] should be replaced by another related to UAE. This one should be placed after MAE.

Lines 54-82: Summarise and focus on chemical/industrial needs.

Line 85: References 10 and 12 should be replaced by 13. These 10 and 12 references should be located in the appropriate place.

Line 87: Reference 13 should be replaced by 14.

Lines 107-123: Summarise and focus on the advantages and disadvantages.

Lines 132-151: Summarise or delete. Focus on what is important for this study: extraction or cultivation?

Lines 169-173: Delete. UAE is a well-known extraction methodology.

Line 184: Delete the text “(see Hinaloc and Roleda)”

Answer: All of the above comments were addressed and updated in the manuscript.

Materials and Methods:

Lines 197-199: Why is important to highlight the difference between the novel variant G-N7 and “Tambalang”? There are no experiments or results regarding Tambalang. Focus on your study or highlight why is important to compare your results to that strain.

Line 216: Delete the text “was received by courier”

Line 218: “It was processed using standard protocols”. Please, clarify/identify the standard protocols applied (and for what) or delete the sentence.

Line 229: Please replace “cheese cloth” by “cloth filter”.

Line 237, 263, 267, 275 …etc: Please check units. The volume of solution added is NOT the ratio volume/mass. It must be always (mL). i.e. 150 mL of NaOH (1M). On the other hand (n=3) can be placed at the end of the sentence or simply say “The extraction was carried out in triplicate”.

Line 237, 303, 313, 318 …etc: Reference [33] refers to a self-study related to Copepods???? Please check carefully and replace for an appropriate one.

Figure 2: Labels are needed for each drawing.

Line 273: The power of the ultrasound bath should be included on lines 277 and 280, not here.

Line 367: References should be placed at the end of the sentence.

Statistical analysis: Have you performed a post-hoc analysis to determine the significant differences among the different extraction methods? It seems impossible that a difference of more than 40% would not show significant differences. I would suggest rechecking all results with Tuckey's test. Maybe the results (and therefore the discussion and conclusions will be different after that).

Line 382: “Figure 6, detailed in Table 1”: Choose one or the other. The table contain much more detailed information.

Line 392: “Figure 6, Table 5”: Choose one or the other.

Line 420. Table 1: Delete “Overview of”

Table 2: Check units. It is “ppm” not “ppm’s”

References: For an experimental article 40-45 references are enough. Please check and delete those not important for this study.

Answer: All of the above comments were addressed and updated in the manuscript.

Reviewer 3 Report

Comments and Suggestions for Authors

The manuscript of Mendes et al. investigates different extraction protocols for the recovery of carrageenans from the red seaweed species Kappaphycus alvarezii. The extraction and subsequent use of seaweeds and seaweed derived products are a widely studied topic that is highly relevant, especially in today’s global situation (as a potential climate-mitigating feedstock). Therefore this paper deserves recognition in this field of research. The approach of the authors is well performed, the article is well written and the applied methodology merits publication in Marine Drugs. However, after reading the manuscript I have some major and minor comments which ought to be addressed before publication:

-          The authors mention following statements in the abstract section: Lines 20-21: “Traditional extraction methods involve alkaline treatment for up to three hours followed by heating, which is inefficient and generates substantial waste.” And further in Lines 30-33: “Additionally, UAE employed 8% KOH to produce semi-refined carrageenan (SRC) from the same biomass, resulting in a similar high yield in half the extraction time and improved carrageenan viscosity, making this technique highly promising for industrial scaling.” It seems that alkali (up to 8%) is still used while applying these advanced extraction technologies. How is substantial waste production avoided compared to the traditional extraction methods?

-          Please capitalize all keywords.

-          Line 186: Ultrasound-assisted extraction and supercritical fluid extraction are known extraction methods, even for the feedstocks applied in this manuscript. Therefore please rephrase.

-          Please specifically emphasize the novelty of this research. Ultrasound-assisted extraction has been performed already numerous times on seaweeds. Therefore, what is the exact novelty of this research? Is it the seaweed species under scrutiny? Process conditions? Please specify.

-          Lines 220-221: what was the moisture content after drying at 60 °C for 48 h?

-          Once an abbreviation is used in the text along its full name, only mention the abbreviation from that point forward.

-          Lines 254-255: please add the details of the drying and grinding protocols.

-          The authors used cheese cloth to filter solutions. Do the authors know the cut off size of the filter material?

-          Please add more details for the UAE subsection, power itself does not say much on its own. What is the amplitude of the UA treatment? Ultrasound intensity is usually depicted as W/cm². What is the intensity of the treatment? Why is only 1 power selected? Surely there are other effects to be observed when changing the power output.

-          For all the treatments different temperatures are chosen (from 70 to 115 °C). How does that allow proper comparison between the different extraction methods?

-          It seems that the protein content is very low according to the Bradford method. Did the authors also consider other (for instance elemental analysis (N) via Dumas) analytical techniques to confirm this?

-          Please improve the quality of Figure 7 and make it more uniform compared to the other figures in the manuscript.

-          Please improve row 1 of Table 3, so text and words are kept together.

-          What is the full biochemical composition of the starting material before extraction?

-          Please add more quantitative statements considering the key experimental results in the conclusions section.

Author Response

Reviewer 3

The manuscript of Mendes et al. investigates different extraction protocols for the recovery of carrageenans from the red seaweed species Kappaphycus alvarezii. The extraction and subsequent use of seaweeds and seaweed derived products are a widely studied topic that is highly relevant, especially in today’s global situation (as a potential climate-mitigating feedstock). Therefore, this paper deserves recognition in this field of research. The approach of the authors is well performed, the article is well written and the applied methodology merits publication in Marine Drugs. However, after reading the manuscript I have some major and minor comments which ought to be addressed before publication:

  1. The authors mention following statements in the abstract section: Lines 20-21: “Traditional extraction methods involve alkaline treatment for up to three hours followed by heating, which is inefficient and generates substantial waste.” And further in Lines 30-33: “Additionally, UAE employed 8% KOH to produce semi-refined carrageenan (SRC) from the same biomass, resulting in a similar high yield in half the extraction time and improved carrageenan viscosity, making this technique highly promising for industrial scaling.” It seems that alkali (up to 8%) is still used while applying these advanced extraction technologies. How is substantial waste production avoided compared to the traditional extraction methods?

Answer: In this study, the traditional extraction time was significantly reduced, and the temperatures were compared to those used in conventional methods. Additionally, the findings suggested that applying concentrations lower than 8% could potentially enhance efficiency.

  1. Please capitalize all keywords.

Answer: We revised as requested.

  1. Line 186: Ultrasound-assisted extraction and supercritical fluid extraction are known extraction methods, even for the feedstocks applied in this manuscript. Therefore, please rephrase.

Answer: We revised as requested

  1. Please specifically emphasize the novelty of this research. Ultrasound-assisted extraction has been performed already numerous times on seaweeds. Therefore, what is the exact novelty of this research? Is it the seaweed species under scrutiny? Process conditions? Please specify.

Answer: We revised the information in the introduction as suggested.

  1. Lines 220-221: what was the moisture content after drying at 60 °C for 48 h? Answer: We added the necessary information on the manuscript.
  2. Once an abbreviation is used in the text along its full name, only mention the abbreviation from that point forward.

Lines 254-255: please add the details of the drying and grinding protocols.

The authors used cheese cloth to filter solutions. Do the authors know the cut off size of the filter material?

Please add more details for the UAE subsection, power itself does not say much on its own. What is the amplitude of the UA treatment? Ultrasound intensity is usually depicted as W/cm². What is the intensity of the treatment? Why is only 1 power selected? Surely there are other effects to be observed when changing the power output.

Answer: We revised the information in the manuscript.

  1. For all the treatments different temperatures are chosen (from 70 to 115 °C). How does that allow proper comparison between the different extraction methods?

Answer: More than comparison, it shows that at lower temperature and time, regarding the UAE and time, regarding SFE, that is possible to achieve similar or higher extraction success.

Answer: We revised the information

  1. It seems that the protein content is very low according to the Bradford method. Did the authors also consider other (for instance elemental analysis (N) via Dumas) analytical techniques to confirm this?

Answer: This is expected for the protein analysis in the carrageenan sample. According to the Codex alimentarius protein needs to be below 2%. The FTIR showed no indication of amide or amine typical bonds associated with the presence of proteins in seaweeds and their extracts.

  1. Please improve the quality of Figure 7 and make it more uniform compared to the other figures in the manuscript.

Answer: We revised the figure accordingly.

  1. Please improve row 1 of Table 3, so text and words are kept together.
  2. Answer: We revised the figure
  3. What is the full biochemical composition of the starting material before extraction? Answer: Although it would have been interesting, no chemical composition of the starting material was performed as the key purpose of these analyses was to understand the purity and composition of the extracted carrageenan for food approval.
  4. Please add more quantitative statements considering the key experimental results in the conclusions section.

Answer: We revised and added more information to the conclusion section as requested.

Round 2

Reviewer 3 Report

Comments and Suggestions for Authors

The authors addressed most of the reviewer's comments and therefore the reviewer accepts this manuscript for publication in Marine Drugs.